# Ovarian Aging: Molecular Mechanisms and Medical Management

**DOI:** 10.3390/ijms22031371

**Published:** 2021-01-29

**Authors:** Jan Tesarik, Maribel Galán-Lázaro, Raquel Mendoza-Tesarik

**Affiliations:** MARGen Clinic, 18006 Granada, Spain; biologas@clinicamargen.com (M.G.-L.); mendozatesarik@gmail.com (R.M.-T.)

**Keywords:** ovarian aging, age-related ovarian decay, premature ovarian insufficiency, genetics of ovarian aging, signaling pathways in ovarian aging, oxidative stress, mitochondrial function, mitochondrial therapy, apoptosis, melatonin, growth hormone

## Abstract

This is a short review of the basic molecular mechanisms of ovarian aging, written with a particular focus on the use of this data to improve the diagnostic and therapeutic protocols both for women affected by physiological (age-related) ovarian decay and for those suffering premature ovarian insufficiency. Ovarian aging has a genetic basis that conditions the ovarian activity via a plethora of cell-signaling pathways that control the functions of different types of cells in the ovary. There are various factors that can influence these pathways so as to reduce their efficiency. Oxidative stress, often related to mitochondrial dysfunction, leading to the apoptosis of ovarian cells, can be at the origin of vicious circles in which the primary cause feeds back other abnormalities, resulting in an overall decline in the ovarian activity and in the quantity and quality of oocytes. The correct diagnosis of the molecular mechanisms involved in ovarian aging can serve to design treatment strategies that can slow down ovarian decay and increase the quantity and quality of oocytes that can be obtained for an in vitro fertilization attempt. The available treatment options include the use of antioxidants, melatonin, growth hormones, and mitochondrial therapies. All of these treatments have to be considered in the context of each couple’s history and current clinical condition, and a customized (patient-tailored) treatment protocol is to be elaborated.

## 1. Introduction

The importance of issues related to ovarian aging has been increasing progressively over the past several decades since, increasingly, more couples in all developed countries choose to postpone parenthood to more advanced female ages [1]. This trend is associated with an increasing rate of aneuploidy in oocytes, causing chromosomal abnormalities in embryos resulting from natural conception, conventional in vitro fertilization (IVF), or intracytoplasmic sperm injection (ICSI) [2,3]. On the other hand, no association has been found between advanced male age and aneuploidy rates in embryos derived from IVF/ICSI attempts using oocytes from young donors [1]. 

The mechanisms involved in ovarian aging are not completely understood and appear to be multifactorial. A better knowledge of the factors and mechanisms causing age-related or premature ovarian decay is needed to optimize diagnostic tests and tune treatment options so as to reflect the individual condition of each couple. This minireview resumes different factors causing ovarian decay and their respective molecular mechanisms of action with the aim of elaborating a patient-tailored treatment regimen for each individual couple.

## 2. Methods

Works used for this minireview were found by searching MEDLINE (PubMed) from 1995 to 2020. A combination of medical subject headings and keywords was used to generate a subset of citations as following (in alphabetical order): (i) antioxidants, (ii) apoptosis, (iii) epigenetic, (iv) genetic, (v) mitochondria, (vi) ovarian aging, (vii) oxidative stress. 

## 3. Molecular Mechanisms

The age of menopause is an inheritable trait, and the age at which primary ovarian failure occurs has a strong genetic component [4,5]. However, not all ovarian failures are primary, and different associated pathologies can play a role. Some of them appear to be related to defective DNA repair pathways [6].

### 3.1. Genetic Basis

Genetic factors can influence ovarian function selectively (primary ovarian insufficiency) or as part of the symptomatology of disorders implicated in other pathologies. 

#### 3.1.1. Primary Ovarian Insufficiency

Primary ovarian insufficiency (POI) is the most frequent cause of early menopause, which occurs in about 10% of women before 45 years of age and in 1–2% before 40 years [4], while fertility impairment starts around 20 years before the menopause [7]. Chromosome X structural abnormalities and X-autosome translocations can be at the origin of some cases of POI [6]. As for single-gene perturbations, several genes have been suggested to be implicated in POI (Table 1), some of them located on the X-chromosome and others on autosomes [4,8,9]. 

The former group includes bone morphogenetic protein 15 (BMP15) (Xp11.2), progesterone receptor membrane component 1 (PGRMC1) (Xq22-q24), androgen receptor (AR) (Xq12), forkhead box O4 (FOXO4) (Xq13.1), premature ovarian failure 1B (POF1B) (Xq21.2), dachshund family transcription factor 2 (DACH2) (Xq21.3), and fragile X mental retardation 1 (FMR1) (Xq27.3). 

The genes on autosomes include growth differentiation factor 9 (GDF9) (5q31.1); folliculogenesis-specific bHLH transcription factor (FIGLA) (2p13.3); newborn ovary homeobox gene (NOBOX) (7q35); nuclear receptor subfamily 5, group A, member 1 (NR5A1); steroidogenic factor-1 (SF-1) (9q33); FSH receptor (FSHR) (2p21-p16); TGF beta receptor III (TGFBR3) (1p33-p32); G protein-coupled receptor 3 (GPR3) (1p36.1-p35); wingless-type MMTV integration site family member 4 (WNT4) (1p36.23-p35.1); inhibins: inhibin alpha (INHA) (2q35), inhibin beta A (INHBA) (7p15-p13), inhibin beta B (INHBB) (2cen-q13); POU class 5 homeobox 1 (POU5F1) (6p21.31); MutS homolog 4 (MSH4) (1p31) and MSH5 (6p21.3); forkhead box O3 (FOXO3) (6q21); cbp/p300-interacting transactivator with Glu/Asp-rich carboxy-terminar domain 2 (CITED2) (6q23.3); spermatogenesis-and oogenesis-specific basic helix-loop-helix transcription factor 1 (SOHLH1) ((9q34.3) and SOHLH2 (13q13.3); phosphatase and tensin homolog (PTEN) (10q23.3); Drosophila nanos homologs 1, 2, and 3: NANOS 1 (10q26.11), NANOS2 (19q13.32), NANOS 3 19p13.13); cyclin-dependent kinase inhibitor 1B (CDKN1B) (12p13.1-p12); anti-Mullerian hormone receptor type II (AMHR2) (12q13); forkhead box O1 (FOXO1) (13q14.1); spalt-like transcription factor 4 (SALL4) (20q13.2); and DNA meiotic recombinase 1 (DMC1) (22q13.1).

However, the implication of some of these genes is suspected rather than confirmed. Those of them with the strongest evidence for playing a role in POI are shown in Table 1.

Most of these genes (Table 1) are known to be somehow related to oogenesis during its different phases, from the fetal period throughout postnatal life until the final phases of meiotic maturation, and to regulate essential events in human oogenesis [4,5]. These include female sex determination and differentiation (*WNT4*), the formation of primordial follicles and the coordinated expression of zona pellucida genes (*FIGLA*), the transition from primordial to growing follicles (*NOBOX, FOXO3, PTEN*), the hormone-dependent phase of follicular growth (*FSHR*), the maintenance of meiotic arrest in antral follicles until the luteinizing hormone (LH) surge (*GPR3*), DNA mismatch repair during meiotic recombination (*MSH4, MSH5*), the apoptosis of ovarian cells (*PGRMC1*), granulosa cell function (*FOXO1*), and the repair of DNA damage arising from defective meiotic divisions (*DMC1*) [4,6,8,9] (Table 1). However, the incidence of anomalies in some of these genes among women suffering from POI is extremely low, and perturbations of other genes are mostly related to specific ethnic groups [10]. Thus, only a few of these genes, such as *FMR1* premutation, *BMP15*, *GDF9*, and *FSHR*, have been incorporated as diagnostic biomarkers [5,11,12]. More research is needed to use more genes as routine diagnostic tools. 

#### 3.1.2. Ovarian Insufficiency due to Mendelian Disorders Implicated in Other Pathologies

Distinct from non-syndromic POI, pleiotropic Mendelian disorders, including fragile X syndrome: familial mental retardation 1 (*FMR1*); (Xq27.3), blepharophimosis-ptosis-epicanthus syndrome (BPES): forkhead box L2 (*FOXL2*) (3q23); galactosemia: galactose-1-phosphate uridyl transferase (*GALT*) (9p13); carbohydrate-deficient glycoprotein syndrome type 1: phosphomannomutase 2 (*PMM2*) (16p13), may manifest POI as part of their phenotypic spectrum [4]. There are some other pleiotropic Mendelian disorders suspected to cause POI, but a definitive confirmation of this association is still lacking.

#### 3.1.3. Gene Mutations Affecting Mitochondrial Function

Genes governing mitochondrial functions may be located in the nucleus or in the mitochondria themselves. Many different nuclear genes affecting mitochondrial function are known nowadays, and according to current experience most gene mutations impairing mitochondrial function are nuclear ones, as reviewed in [4]. Both oocyte and ovarian cell mitochondria are important for the correct folliculogenesis and oocyte maturation [13,14,15,16]. As to the risk of mitochondrial DNA deletions, it was shown to be affected by the global secondary structure of the mitochondrial genome [17]. 

### 3.2. Cell-Signaling Pathways

The main cell-signaling pathways involved in physiological (age-related) ovarian failure and POI are those involved in cell protection against oxidative stress. A recent study using single-cell transcriptomic analysis of ovaries from young and aged non-human primates identified seven ovarian cell types with distinct gene-expression signatures, including oocyte and six different somatic cells, and identified the disturbance of antioxidant signaling specific to early-stage oocytes and granulosa cells [9]. The further analysis of cell-type-specific aging-associated transcriptional changes uncovered age-related disturbances of antioxidant signaling specific to early-stage oocytes and granulosa cells [9]. The authors have completed their observations on non-human primates with those on human granulosa cells obtained from follicular fluid samples aspirated from patients undergoing an IVF attempt. Consistent with the results in monkeys, human granulosa cells exhibited the age-related downregulation of the transcription of three genes involved in antioxidative pathways—*IDH1*, *PRDX4*, and *NDUFB10*—and this phenomenon was accompanied by an increase in reactive oxygen species and apoptosis in the granulosa cells [18], indicative of oxidative damage as a crucial factor in ovarian functional decay. 

Similar to the primates, an age-related increase in oxidative damage and a decrease in antioxidant gene expression was previously observed in mice [14]. The oxidative damage of mouse granulosa cells was reported to impair the supply of ATP and the mitochondrial gene expression, which are required not only for the proliferation but also the differentiation of granulosa cells during follicular development [13]. Moreover, aging-related mitochondrial (mt) DNA instability also leads to an accumulation of mtDNA mutations in the oocyte, leading to the deterioration of oocyte quality in terms of competence and the risk of transmitting mitochondrial abnormalities to offspring [14]. As mentioned above, the risk of mtDNA damage is also conditioned by epigenetic factors, especially the global secondary structure of the mitochondrial genome; certain patterns of the global secondary structure of the human single-stranded heavy chain of mtDNA make the DNA molecule more prone to deletions than others [17]. 

Mitochondrial damage leads to the activation of apoptotic pathways in granulosa cells, which, in turn, decreases the expression of aromatase, which is required for the transformation of androgens to estrogens, thus leading to the prevalence of androgens over estrogens within the follicles [18]. Androgens and estrogens present in follicular fluid exert rapid non-genomic effects on maturing human oocytes, affecting oocyte cytoplasmic maturation and postfertilization developmental potential rather than the completion of meiosis [19]. This can explain why age-related or premature disturbances of antioxidative pathways reduce oocyte quality even in cases in which the number of mature oocytes recovered for IVF is not affected. 

New data suggest that POI may be of polygenic origin, and that overlap exists between the genetic backgrounds of diminished ovarian reserves and POI [20]. Whole-exome sequencing and bioinformatics analysis may become a useful clinical tool for etiological diagnosis and risk prediction for affected women in the future [20]. In addition, array comparative genomic hybridization or specific next-generation sequencing panels should be considered to identify chromosomal deletions/duplications under karyotype resolution or other pathogenic variants in specific genes associated with POI. This is particularly important in patients with first-or second-degree relatives also affected with POI, improving their reproductive and genetic counseling [21].

Moreover, even with the same genetic background, with the use of multiple systems biologic approaches to compare developmental stages in the early human embryo with single-cell transcript data from blastomeres, it was shown that blastomeres considered to be totipotent were not transcriptionally equivalent [22]. 

## 4. Clinical Management

### 4.1. Diagnosis

A woman’s age is the basic predictor of the degree of ovarian aging. However, ovarian aging can develop prematurely, and specific diagnostic methods are needed to detect this condition. There are two types of ovarian aging manifestations with respect to the ovarian ability to produce oocytes: a quantitative one and a qualitative one. To detect the former, a combination of antral follicle count (AFC), determined by vaginal ultrasound scan at the beginning of the menstrual cycle, and the determination of serum anti-Mullerian hormone (AMH) concentration, which can be performed at any time during the menstrual cycle [23], is used.

By contrast, oocyte quality is not necessarily related to AFC and serum AMH. Premature decay of oocyte quality is mainly related to a low-for-age production of growth hormone (GH) [24]. Due to the pulsatile pattern of GH secretion, insulin-like growth factor-1 (IGF-1), which does reflect the GH secretion pattern, but with a less pronounced pulsatility, was suggested to identify young patients with a premature decay of oocyte quality who could benefit from treatment with GH during ovarian stimulation [25]. The data presented show that the ‘GH/IGF-1’ age can be more than 20 years higher than the chronological age in some young women, and it was this group of patients who appeared to be more likely to benefit from GH administration during ovarian stimulation, although the authors suggested that larger prospective studies were needed to confirm this assumption [25].

There are only a few clues to detect premature ovarian aging in addition to the above criteria, and further research is warranted to detect more molecular markers that could guide the clinician to propose the best treatment regimen.

### 4.2. Treatment

In view of the fact that oxidative stress is the main factor involved in ovarian aging, it can be assumed that agents reducing oxidative damage represent the first-line choice. These agents can be direct antioxidants or molecules affecting cell-signaling pathways involved in the antioxidant defense of ovarian cells (Table 2). Some molecules combine both of the above activities.

#### 4.2.1. GH

GH administration during ovarian stimulation was the first treatment shown to be beneficial in older women. A randomized controlled trial, conducted in a group of 100 women of >40 years undergoing assisted reproduction treatment and randomized between a GH treatment group and a placebo group, showed significantly higher delivery and live birth rates in the GH arm as compared to the placebo arm [26]. Subsequent studies confirmed these findings and extended the use of GH treatment also to younger women with POI [27,28,29,30,31]. This effect of GH is at least partly due to the alleviation of oxidative stress in the ovary, an effect previously described in some other organs [25]. Though it is not a direct antioxidant, GH intervenes in the cell-signaling pathways involved in cellular defense against oxidative stress [28], and adult GH deficiency causes an inadequate reactivity of cells against radical production [29]. This can explain why age-related or premature GH deficiency contributes to ovarian decay, even in cases where it is primarily due to other causes. Hence, GH can be used as an adjuvant treatment during ovarian stimulation in women with both age-related ovarian decay and POI. This conclusion was drawn from the results of 13 articles of a special research topic on the role of GH in reproduction, edited by Jan Tesarik, John Yovich, and Yves Menezo, and published in *Frontiers in Endocrinology*, reviewed in Tesarik et al. in 2021 [32].

#### 4.2.2. Melatonin

Melatonin is an example par excellence of a molecule acting as a direct antioxidant and a modulator of systems protecting cells against oxidative stress. First introduced into reproductive medicine to treat infertility caused by endometriosis and adenomyosis [33,34], it has proven its usefulness in many more indications, including the protection against the contraction of COVID-19 [35]. Since then, new data have emerged showing the possibilities of melatonin in preventing ovarian aging [36]. Even though the mechanism underlying the anti-aging effect of melatonin in the human ovary still needs to be fully explained, the administration of melatonin, lacking serious side-effects and providing additional benefits to patients treated with it [36], is clearly indicated in women with age-related ovarian decay or POI.

#### 4.2.3. Other Antioxidants

Since ovarian aging is mainly due to oxidative stress, superimposed on the existing genetic makeup, any antioxidant agent can be supposed to improve IVF results in older women and in young women suffering from POI [37,38,39]. Unfortunately, there are only a few clinical studies that can conclusively support this reasoning. Coenzyme Q10 (CoQ10) is the antioxidant that accumulates the most evidence in favor of its use in the treatment of women with diminished ovarian reserves [37,38], a conclusion corroborated by a recent randomized control trial [39]. However, though potentially less efficient, any antioxidant agent, such as vitamins C and E and folic acid, can be of help [40]. 

#### 4.2.4. Mitochondrial Therapy

The issue of mitochondrial health has been widely debated with respect to the oocyte [41]. Nevertheless, data obtained from a study in an animal model (mouse) indicate that the granulosa cell mitochondria are no less important for the correct oocyte maturation than those of the oocyte itself [15]. Antioxidant agents, such as melatonin, coenzyme Q10, or vitamins C and E (see above) can improve this condition. This can explain why antioxidant pretreatment (coenzyme Q10) improves the ovarian response to gonadotropin stimulation and embryo quality in low-prognosis young women [37,38,39].

On the other hand, the accumulation of mitochondrial DNA mutations/deletions in the oocyte, in addition to jeopardizing oocyte developmental potential, can also compromise the health of the offspring [42]. Optimal mitochondrial function is required for oocyte maturation, fertilization, and embryonic development. Improvement of the mitochondrial function, either through the use of small molecules or procedures involving mitochondrial transfers, could lead to better fertility outcomes. Moreover, mitochondrial replacement procedures could open a new page in the treatment of mitochondrial diseases.

Mitochondrial transfer from a healthy donor oocyte to the patient’s oocyte is a possible solution, practiced from the late 1990s [42,43,44,45] but thereafter banned in most countries. This technique can be performed in two different ways: first, by injection of a small amount of donor oocyte cytoplasm into the patient’s oocytes [42,43,44]; second, by transferring the metaphase chromosomes, associated with the meiotic spindle, from the patient’s oocytes to previously enucleated donor oocytes [45]. Curiously, even though the latter technique results in a much higher proportion of “healthy” mitochondria in the reconstructed oocytes as compared to the former one, the efficiency of both techniques appeared similar; more than 40% of live births in young women with previous implantation failures, and several tens of births were obtained with both of them [46,47]. It has to be stressed, however, that the original indication of both of these techniques was focused on the recurrent failure of embryo development and implantation in young women, rather than on the avoidance of mitochondrial disease transmission. In addition to mitochondria, ooplasm also contains other developmentally important molecules that can be deficient in patients’ oocytes. One of them is stored maternal mRNA which is crucial for guiding human embryo development until the 4-to 8-cell stage, when the first signs of embryonic genome expression can be detected [48,49,50,51]. However, stored maternal mRNA is also involved in the control of relatively late stages of human preimplantation development, after the activation of embryonic gene expression, when the two mRNA sources act together to regulate the differentiation of the first two embryonic tissues, the inner cell mass, and the trophectoderm [51]. 

While still banned in the United States, the technique of mitochondrial replacement therapy, using a spindle-chromosome complex transfer from the patient’s oocytes to enucleated donor oocytes, was used with success by a team in a US clinic in Mexico, where there is no legal restriction with regard to this technique, to avoid the mother-to-offspring transmission of a heritable mitochondrial disease, the Leigh syndrome [52]. The authors chose the previously described technique of nuclear transfer, with slight modifications [53], rather than ooplasmic injection. This technique is currently used, in countries in which it is legally possible, for its initial indication: the repeated failure of embryonic development in young couples with male partners having normal spermatozoa, without strict limitations to previously detected mitochondrial DNA abnormalities.

#### 4.2.5. Patient-Tailored (Customized) Treatment Protocols

In order to be able to decide between different treatment options, the choice of the therapy to be used must be made on the basis of a complete diagnosis of both the male and female partner of each couple. The treatment choice should not be based merely on the primary cause of infertility, but should also take into account all possible secondary contributing factors. Basic guidelines for this approach were published under the name “CARE” (Customized Assisted Reproduction Enhancement) [54]. In bad-prognosis patients, this individualized approach seems to be more efficient as compared with standard protocols [55]. Further prospective studies are needed to definitively confirm this thesis.

## 5. Conclusions

All the available data lead to the conclusion that ovarian aging is mainly due to oxidative stress. While the external factors leading to oxidative stress might be similar, the damage produced is patient-dependent due to genetically programmed defense mechanisms. Deficiencies of these mechanisms can be caused by mutations/deletions of both nuclear and mitochondrial genes. Independent of the genetic background (which is impossible to resolve at present), the clinical manifestations can be alleviated with treatments using direct antioxidants (e.g., vitamins C and E, coenzyme Q10), agents affecting the cell response against oxidative stress (e.g., growth hormone), or those combining both of these activities (e.g., melatonin). If mitochondrial DNA mutations occur in the oocyte, mitochondrial replacement therapy can resolve the problem. Independent of the results of the individual diagnostic tests performed, a synthetic view is needed to propose a customized therapeutic plan for each infertile couple.

## Figures and Tables

**Table 1 ijms-22-01371-t001:** Overview of the principal nuclear genes with a known role in the protection of the ovaries against aging ^1^.

Gene	Location	Function
*WNT4*	1p36.23-p35.1	Female sex determination and differentiation
*FIGLA*	2p13.3	Primordial follicle and zona pellucida formation
*NOBOX*	7q35	Transition from primordial to growing follicles
*FOXO3*	6q21	Transition from primordial to growing follicles
*PTEN*	10q23.3	Transition from primordial to growing follicles
*FSHR*	2p21-p16	Hormone-dependent phase of follicular growth
*GPR3*	1p36.1-p35	Maintenance of meiotic arrest until the LH surge
*MSH4*	1p31	DNA mismatch repair during meiotic recombination
*MSH5*	6p21.3	DNA mismatch repair during meiotic recombination
*PGRMC1*	Xq22-q24	Apoptosis of ovarian cells
*FOXO1*	13q14.1	Granulosa cell function
*DMC1*	22q13.1	Repair of DNA damage during meiotic divisions

^1^ See the main text for the full name of each gene. Full review of the genes can be found in references [4,8,9].

**Table 2 ijms-22-01371-t002:** Agents that can be used to treat the consequences of physiological or premature ovarian aging.

Agent	Administration	Mechanisms of Action	References
GH	Subcutaneous	Activation of cell-signaling pathways acting	[26,27,28,29,30,31,32]
		against oxidative stress	
		Possible activation of DNA damage repair	
Melatonin	Oral	Direct antioxidant	[33,34,35,36]
		Indirect antioxidant (signaling pathway modulator)	
		Anti-inflammatory agent	
		Immunomodulator	
Coenzyme Q10	Oral	Direct antioxidant	[37,38,39]
Vitamin C	Oral	Direct antioxidant	[40]
Vitamin E	Oral	Direct antioxidant	[40]
Folic acid	Oral	Direct antioxidant	[40]

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
