# Peer review of "Ovarian Aging: Molecular Mechanisms and Medical Management"

_ijms, 2021, doi:10.3390/ijms22031371_

Round 1

Reviewer 1 Report

Dear author,

The review focuses on molecular mechanisms and medical management of ovarian aging, mainly to improve diagnostic and therapeutical protocols for women. The manuscript is well written and structures, but some changes are necessary. First, authors should include some more recent work related to ovarian aging, related to molecular mechanisms, such as Ma et al. (2020), or Turan and Oktay (2020), between others. In addition, they should include current specific treatments and their concrete results. In order to improve presentation, authors should justify all text. Other changes required are:

  • Line 19: “in vitro” must be written in italics.
  • Line 77: incorrect reference.
  • Tables 1 and 2: add references.
  • Lines 105-107: add references that support this sentence.
  • Lines 108-113: add references that support this sentence.
  • Line 114: old review. Authors could include more up-to-date information.
  • Line 177: wrongly indicated reference.
  • Lines 247-249: this information is not relevant to subject of review. Authors should include information about other indications for melatonin, mainly related to reproduction.
  • Line 249: wrongly indicated reference.
  • Lines 277-278: add reference that support this sentence.
  • Lines 283-284: authors should include information about results obtained with these techniques.
  • Line 290: eliminate one “the”.

Author Response

Responses to Reviewer 1 Comments

  1. The review focuses on molecular mechanisms and medical management of ovarian aging, mainly to improve diagnostic and therapeutical protocols for women. The manuscript is well written and structures, but some changes are necessary. First, authors should include some more recent work related to ovarian aging, related to molecular mechanisms, such as Ma et al. (2020), or Turan and Oktay (2020), between others. In addition, they should include current specific treatments and their concrete results. In order to improve presentation, authors should justify all text. Other changes required are:

Response: We have added the suggested references and some more related text to the revised manuscript.

  • Line 19: “in vitro” must be written in italics.

Response: “”in vitro has been written in italics

  • Line 77: incorrect reference.

Response : We have corrected this reference.

  • Tables 1 and 2: add references.

Reponse: We have added detailed references for each treatment in Table 2. In Table 1, we have added 3 review articles containing detailed references for each gene, as a  footnote. I hope this will be OK. The data about each gene are so complete in those references, in a tabulated form, that publishig these tables here again might be close to plagiarism.

  • Lines 105-107: add references that support this sentence.

Response: Two references have been added here.

  • Lines 108-113: add references that support this sentence.

Response: Four references have been added here.

  • Line 114: old review. Authors could include more up-to-date information.

Response: We have added a reference from 2020 in this place.

  • Line 177: wrongly indicated reference.

Response: We have corrected the error.

  • Lines 247-249: this information is not relevant to subject of review. Authors should include information about other indications for melatonin, mainly related to reproduction.

Response: We have added one recent citatiion, related to repoduction, and removed another  one, related to COVID-19. However, I consider that preventive use of melatonin against  COVID-19 is also important because nobody can predict what long-term effects COVID-19 can produce in the ovaries and if these effects could reduce ovarian reserve.

  • Line 249: wrongly indicated reference.

Response: We have removed reference 33.

  • Lines 277-278: add reference that support this sentence.

Response: We have added the corresponing references

  • Lines 283-284: authors should include information about results obtained with these techniques.

Response: The livebirth rate was over 40% in young women with previous implantation failures. This sentence has been added to the revised manuscript.

  • Line 290: eliminate one “the”.

Response: The typographical error has been correcred.

Reviewer 2 Report

The work presented for review is valuable, it is an important novelty, but it omits many valuable topics, e.g. : the oxidative damage hypothesis. 

There is no subsection "Materials and Methods" in which the authors should describe the qualification criteria for works taken for this review.

Table 1 and table 2 are without references.

provide an explanation of the abbreviation only the first time you use it, e.g. growth hormone (GH).

Author Response

The work presented for review is valuable, it is an important novelty, but it omits many valuable topics, e.g. : the oxidative damage hypothesis. 

Response: The oxidative damage hypothesis is a key-idea of the manuscript, and both direct and indirect antioxidants are suggested as principal therapeutic agents (Table 2). We have highlighted the sentences underscoring the importance of oxidative stress in the Abstract, in the Keywords and in the Conclusions  of the revised manuscript.

There is no subsection "Materials and Methods" in which the authors should describe the qualification criteria for works taken for this review.

Response: We have added a “Methods” subsection where the qualification criteria for works included in this minireview are described.

Table 1 and table 2 are without references.

Reponse: We have added detailed references for each treatment in Table 2. In Table 1, we have added 3 review articles containing detailed references for each gene, as a footnote. I hope this will be OK. The data about each gene are so complete in those references, in a tabulated form, that publishing these tables here again might be close to plagiarism.

provide an explanation of the abbreviation only the first time you use it, e.g. growth hormone (GH).

Response: We have done the suggested modification throughout the revised manuscript.

Round 2

Reviewer 1 Report

Dear authors,

The manuscript has markedly improved its quality. Thanks for including my recommendations.